# How CD4^+^ T Cells Transcriptional Profile Is Affected by Culture Conditions: Towards the Design of Optimal In Vitro HIV Reactivation Assays

**DOI:** 10.3390/biomedicines11030888

**Published:** 2023-03-13

**Authors:** Giuseppe Rubens Pascucci, Elena Morrocchi, Chiara Pighi, Arianna Rotili, Alessia Neri, Chiara Medri, Giulio Olivieri, Marco Sanna, Gianmarco Rasi, Deborah Persaud, Ann Chahroudi, Mathias Lichterfeld, Eleni Nastouli, Caterina Cancrini, Donato Amodio, Paolo Rossi, Nicola Cotugno, Paolo Palma

**Affiliations:** 1Research Unit of Clinical Immunology and Vaccinology, Bambino Gesù Children’s Hospital, 00165 Rome, Italy; 2Department of Systems Medicine, University of Rome “Tor Vergata”, 00133 Rome, Italy; 3Department of Pediatric Infectious Diseases, School of Medicine, Johns Hopkins University, Baltimore, MD 21218, USA; 4Department of Pediatrics, School of Medicine, Emory University, Atlanta, GA 30322, USA; 5Center for Childhood Infections and Vaccines, Children’s Healthcare of Atlanta and Emory University, Atlanta, GA 30322, USA; 6Ragon Institute of MGH, MIT and Harvard, Cambridge, MA 02139, USA; 7Infectious Disease Division, Brigham and Women’s Hospital, Boston, MA 02115, USA; 8Department of Infection, Immunity and Inflammation, UCL Great Ormond Street Institute of Child Health, London WC1N 1EH, UK

**Keywords:** T cell activation, in vitro cultures, HIV reactivation, RNA sequencing, Oxford Nanopore technologies, autologous plasma, TCR signaling cascade, PMA/ionomycin stimulation, CD4^+^ T cells, RPMI

## Abstract

Most of the current assays directed at the investigation of HIV reactivation are based on cultures of infected cells such as Peripheral Blood Mononuclear Cells (PBMCs) or isolated CD4^+^ T cells, stimulated in vitro with different activator molecules. The culture media in these in vitro tests lack many age- and donor-specific immunomodulatory components normally found within the autologous plasma. This triggered our interest in understanding the impact that different matrices and cell types have on T cell transcriptional profiles following in vitro culture and stimulation. Methods: Unstimulated or stimulated CD4^+^ T cells of three young adults with perinatal HIV-infection were isolated from PBMCs before or after culture in RPMI medium or autologous plasma. Transcriptomes were sequenced using Oxford Nanopore technologies. Results: Transcriptional profiles revealed the activation of similar pathways upon stimulation in both media with a higher magnitude of TCR cascade activation in CD4^+^ lymphocytes cultured in RPMI. Conclusions: These results suggest that for studies aiming at quantifying the magnitude of biological mechanisms under T cell activation, the autologous plasma could better approximate the in vivo environment. Conversely, if the study aims at defining qualitative aspects, then RPMI culture could provide more evident results.

## 1. Introduction

The ability of HIV to integrate into the human genome and persist in a latent phase within long-lived host cells represents the main obstacle to cure. Among people living with HIV and treated with ART (antiretroviral therapy), integrated viral DNA can be found mainly in resting CD4^+^ T cells [1,2]. More than 90% of this latent reservoir consists of defective proviral genomes that can contribute to chronic immune activation and exhaustion but are not responsible for viral rebound if ART is interrupted [3]. The inducible or replication-competent proviruses, the key targets of curative efforts, can either be activated naturally (at ART discontinuation and/or suboptimal adherence) or in vitro with specific stimulation approaches. Assays such as the Quantitative Viral Outgrowth Assay (QVOA) and Tat/rev Limiting Dilution Assay (TILDA), that have been developed to measure the replication-competent or inducible reservoir, require in vitro stimulation of CD4^+^ T cells isolated from ART-suppressed individuals with HIV. However, these protocols mostly use RPMI-based media that not only contain animal-derived products [4,5], but are also not representative of the complexity of human plasma, a rich source of age-specific immunomodulatory components such as antibodies, complement, cytokines, chemokines, and metabolites [6]. In this context, we tried to determine the impact of selecting this artificial medium, largely employed in research, in comparison to human plasma and how the choice of culturing and stimulating a single lymphocyte population (CD4^+^ T cells) as opposed to bulk PBMCs influences CD4^+^ T cell gene expression profiles.

## 2. Materials and Methods

### 2.1. Ethics Statement

The study was conducted in accordance with the ethical guidelines approved forth by the Institutional Ethics Committee of Bambino Gesù Children’s Hospital on 16 July 2021 with protocol number 2555_OPBG_2021. All participants or their legal guardians gave written informed consent to the study in accordance with the Declaration of Helsinki.

### 2.2. Study Participants

Three young adults living with perinatally-acquired HIV infection (two female and one male) (S1, S2, S3) (age range 12–26 years), who started ART within the first year of life and with a history of suppressive ART > 2 years, were recruited at Bambino Gesù Children’s Hospital (Rome, Italy). Inclusion criteria were the following: age > 12 years; age at ART start < 1 year; and history of suppressive ART > 2 years. TILDA revealed an inducible, transcriptionally competent HIV reservoir in subject S3, but not S1 and S2. Furthermore, total HIV DNA load in CD4^+^ T cells was quantified for all the individuals and was 255, 35, and 377 HIV-1 DNA copies per million CD4^+^ T cells for S1, S2, and S3, respectively. Cell-associated HIV RNA was estimated for Long Terminal Repeats (LTRs) (0, 45.6, and 0 RNA copies per million PBMCs, respectively) and for pol gene (0.64, 0, and 0 RNA copies per million PBMCs, respectively).

### 2.3. Sample Collection

Blood samples were obtained by blood draws collected in EDTA (Ethylenediaminetetraacetic acid) tubes and 1 mL of plasma was aliquoted after centrifugation and stored in cryovials at −80 °C until further analysis. Peripheral Blood Mononuclear Cells (PBMCs) were isolated using Ficoll-Paque PLUS (Code#: 17144003, Cytiva Sweden AB, SE-751 84 Uppsala, Sweden) density gradient centrifugation. Countess II Automated Cell Counters (ThermoFisher Scientific, Waltham, MA, USA) was used to determine the total number of cells, then PBMCs were frozen and stored in Liquid Nitrogen at the Cryolab facility placed at the University of Rome “Tor Vergata”.

### 2.4. Study Design and In Vitro Stimulation

CD4^+^ T cells of ART-treated young adults with perinatal HIV-infection were enriched by negative selection with the EasySep Human CD4^+^ T Cell Enrichment Kit (Cat. #: 19052, Stemcell Technologies, Vancouver, BC, Canada) according to manufacturer’s instructions from thawed PBMCs before (pre-culture condition, PREc) or after (post-culture condition, POSTc) the 18 h in vitro stimulation. An aliquot of non-cultured CD4^+^ T cells (T0) was collected as a control sample to characterize the transcriptome profile of CD4^+^ T before the in vitro culture. CD4^+^ T cells (in PREc) or PBMCs (in POSTc) were cultured in round-bottom 96-well plates (≤1 × 10^6^ cells/well) for 18 h (at 37 °C, 5% CO_2_) and activated with 81 nM Phorbol 12-myristate 13-acetate (PMA, cat. #: P1585, Sigma-Aldrich, Merck KGaA, Darmstadt, Germany), 1,34 uM IONOMYCIN (IONO, cat. #: I0634, Sigma-Aldrich, Merck KGaA, Darmstadt, Germany), 20 U/uL recombinant human Interleukin-2 (IL-2, cat. #: 200-02, Peprotech, Cranbury, NJ, USA), and the HIV-1 reverse transcriptase inhibitor Zidovudine (formerly Azidothymidine, AZT, cat. #: 3485 from National Institutes of Health, NIH, HIV Research Program, HRP, Bethesda, MD, USA) used at 400 nM to avoid in vitro HIV re-infection (ST) or left unstimulated (US). Cells were both resuspended in autologous plasma without supplements or 1640 RPMI (Cat. #: ECM9106L, EuroClone, Milan, Italy) supplemented with 10% Fetal Bovine Serum (FBS, ref. #: 10270-106, Gibco, ThermoFisher Scientific, Waltham, MA, USA), 1% L-Glutamine (Cat. #: ECB3000D, EuroClone, Milan, Italy), and 1% Penicillin/Streptomycin solution (Cat. #ECB3001D, EuroClone, Milan, Italy), to later carry out RNA-seq using Oxford Nanopore technology. Live cells were measured with Countess II Automated Cell Counters (ThermoFisher Scientific, Waltham, MA, USA) for each condition at the end of the cell culture. Unstimulated samples had a viability ≥80%, while stimulated samples had a viability between 70 and 80%.

### 2.5. RNA Isolation

CD4^+^ T cells were first lysed in a solution of RNeasy Lysis Buffer (RLT Buffer, QIAGEN, Venlo, The Netherlands) and beta-mercaptoethanol (BME, ref. #: 21985-023, Gibco, ThermoFisher Scientific, Waltham, MA, USA, using 10 μL BME for each mL of RLT) and then stored at −20° overnight for RNA yield maximization. The day after, according to cell number availability, RNA isolation was made via the RNeasy™ Mini and Micro Kit™ (Cat. #: 80204 and #: 80284 respectively, QIAGEN, Venlo, The Netherlands) following manufacturers’ instructions. Pure, concentrated RNA was eluted in 14 μL (Micro kit) or 30 μL (Mini kit) RNase-free water (Mat. No. #: 1039480, QIAGEN, Venlo, The Netherlands) and 1 μL of RNAse Inhibitor (Cat. #: AM2694, Invitrogen, ThermoFisher Scientific, Waltham, MA, USA) was added to avoid sample degradation. The RNA concentration (ng/μL sample) and purity (absorbance ratios A260/280 and A260/230) was measured by Nanodrop one (ThermoFisher Scientific, Waltham, MA, USA) using 1 μL/sample. At the end of the procedure, the isolated RNA was stored at −80 °C until further analysis.

### 2.6. RNA Sequencing

Bulk RNA sequencing was performed using the long-reads Oxford Nanopore Technologies (Oxford, UK). The PCR-cDNA Sequencing Kit SQK-PCS109 was used for S1 PREc samples; meanwhile, the PCR-cDNA Barcoding kit SQK-PCB109 was used to multiplex the samples of each donor in a single library. The sequencing libraries were prepared following the manufacturer’s protocol. In brief, starting from 50 ng of full-length RNA, complementary strand DNA (cDNA) synthesis and strand switching were performed using kit-supplied oligonucleotides. dscDNA was generated by PCR amplification of cDNA (10 μL cDNA/sample) using primers with 5′ tags which facilitate the ligase-free attachment of Rapid Sequencing Adapters. The prepared libraries were either immediately used for loading onto the R9.4.1 flow cell (FLO-MIN106) and sequenced or stored at −80 °C until use. Libraries were run on MinION flow cells with MinKNOW acquisition software version v.21.02.01.

### 2.7. HIV-1 DNA Quantification on CD4 T Cells by Droplet Digital PCR

HIV-1 DNA levels in purified CD4^+^ T cells were measured by the QX200™ Droplet Digital™ PCR (ddPCR) system (Bio-Rad, Pleasanton, CA, USA). Briefly, ddPCR mix was prepared mixing isolated DNA with 2X ddPCR Supermix for Probes; LTRfw and LTRrv primers [7], or hTERTfw and hTERTrv primers [8], and LTR or hTERT (human telomerase reverse transcriptase) probe, respectively. QX200™ Droplet Generator (Bio-Rad, Pleasanton, CA, USA) was used to generate droplets, then the plate was placed into a 2720 Thermal Cycler (Applied Biosystems, ThermoFisher Scientific, Waltham, MA, USA) with the following cycling conditions: 94 °C for 10 min; 45 cycles at 94 °C for 30 s and 58.5 °C for 1 min; and 98 °C for 10 min. The droplets were then read by the QX200™ Droplet Reader (Bio-Rad, Pleasanton, CA, USA) and the results were later analyzed with the QuantaSoft™ Analysis Software v.1.7.4.0917 (Bio-Rad, Pleasanton, CA, USA). Only wells with ≥10,000 droplets were considered for the analysis. Each sample was run in triplicate. The HIV-1 copy number was normalized to the hTERT copy number (diploid gene), and the results were expressed as HIV-1 DNA copies/10^6^ CD4 cells.

### 2.8. Quantitative CA-HIV-1 RNA Assay (qRT-PCR)

Cell associated HIV-1 RNA (CA-HIV-1 RNA) was quantified as previously described [9]. Briefly, total cellular RNA was extracted using the automated Qiasymphony platform (DSP virus/pathogen mini kit, QIAGEN, Venlo, The Netherlands). Similarly, RNA was processed in-house to selectively amplify total (LTR) and unspliced (pol) ca-HIV-1 RNA by qRT-PCR applying primers from previously validated assays [7,10]. The caHIV-1 RNA measurements were standardized against the expression of the cellular genes TBP1 and IPO8 to determine the number of copies of caHIV-1 RNA per 106 PBMCs.

### 2.9. Data Analysis

#### 2.9.1. RNA-Seq Raw Data Processing

Nanopore FAST5 files were basecalled with the stand-alone Guppy basecaller v.6.2.11. In not-multiplexed samples (S1 PREc samples), adapters were trimmed during basecalling. For the other samples, adapters and barcodes were trimmed during demultiplexing that was performed using Guppy barcoder v6.2.11. The quality of the reads was checked with the fastqc tool v.0.11.9 and the reads with more than 40% of the base with a Phred score of less than 10 were filtered out with fastp v.0.12.4. Next, the high-quality reads were aligned against hg38 Gencode human transcriptome reference v38 using minimap2 v.2.21 with map-ont preset parameters. Reads unmapped on hg38 transcriptome were aligned against the Human immunodeficiency virus type 1 (HXB2) genome sequence (GenBank: K03455.1) using minimap2 with the splice preset parameters. SAM-to-BAM conversion, BAM sorting, indexing, and extraction of alignment statistics was performed with samtools v1.10. Salmon v0.13.1 with “--gcBias -l U” parameters was used to summarize the mapped reads. Transcript level estimates were aggregated to gene level raw counts using the tximport R package v.1.22.0. Raw count values were then transformed using the edgeR v.3.36.0 R package into normalized counts per million (CPM) values. Genes with a mean of CPM under 5 were excluded.

#### 2.9.2. Principal Component Analysis

Principal Component Analyses (PCA) of sample expression levels were performed using the prcomp function from the R package stats v.4.1.1 scaling and centering the CPM values.

#### 2.9.3. Differential Expression Analysis

To identify differentially expressed genes (DEGs) between conditions, we decided to analyze donors individually instead of using them as biological replicates. The edgeR package, applying an exact test coupled with the negative binomial distribution, can also perform statistical tests with a single replicate if the Biological Coefficient of Variation (BCV) is provided. Typically, this value is 0.4 for well-controlled experiments of human samples, 0.1 for samples of genetically identical model organisms, or 0.01 for technical replicates [11]. Considering that we compared conditions within each donor and not among donors, we set the BCV parameter at 0.3. Among all possible comparisons, we tested the 20 most informative and we selected genes with an adjusted *p*-value (FDR) (False discovery rate) < 0.05 and a |logFC| > 1.4.

#### 2.9.4. Gene Set Enrichment Analysis

Gene set enrichment analysis (GSEA) was performed using the gseGO function in the clusterProfiler (v. 4.2.2) [12] R package and the Biological Processes gene sets. For all the 20 comparisons, genes were ranked according to the log of the fold change multiplied by -log of FDR. A normalized enrichment score (NES) is calculated for each Biological Process. This NES reflects the degree to which a gene set is over-represented at the top (UP genes) or bottom (DW genes) of a ranked list of genes. In other words, it can highlight whether a gene set is enriched in our deregulated genes.

#### 2.9.5. Virtual Inference of Protein-Activity by Enriched Regulon Analysis (VIPER)

DecoupleR (v. 2.0.1) [13] R package was used to investigate the activity of transcription factors (TF) and pathways through VIPER (Virtual Inference of Protein-activity by Enriched Regulon analysis) algorithm. We coupled VIPER with the DoRothEA (Discriminant Regulon Expression Analysis) [14] to infer TFs (transcription factors) activity and with the PROGENy (Pathway RespOnsive GENes for activity inference from gene expression) [15] databases for pathways activity estimation. VIPER allows computational inference of protein activity in single-sample gene expression profiles or using the output of a differential analysis [13]. We used the first strategy to highlight similarities and differences between individual samples. Taking into account the low differences observed between the PREc and POSTc isolation strategies and to reduce the number of comparisons, we aggregated the normalized counts of the PREc and POSTc samples by mean and performed a new differential analysis (Appendix A). We used the ranking indices from this analysis to focus on the differences between the conditions.

#### 2.9.6. Differential Analysis of Genes Downstream the TCR Signaling Cascade

We compared the expression levels of some genes regulated by key TFs downstream of TCR (T cell receptor) signaling. Statistical comparisons between unstimulated plasma (PLSM US), stimulated plasma (PLSM ST), unstimulated RPMI (RPMI US), and stimulated RPMI (RPMI ST) were performed using the Kruskal–Wallis test followed by Dunn’s post hoc test. Post hoc tests were applied only to the genes with a *p*-value adjusted (FDR) <0.05.

## 3. Results

### 3.1. Overview of Transcriptional Profiles in Different Conditions

We explored the full transcriptome of CD4^+^ T cells from three ART-suppressed donors with perinatal HIV, evaluating the impact of isolating the CD4^+^ T cells pre-culture (PREc) or post-culture (POSTc), using autologous plasma (PLSM) or RPMI as the culture matrix. Cells were stimulated (ST) for 18 h with PMA/ionomycin and IL-2 together with AZT or left unstimulated (US) for both matrices (Figure 1). Taking into account all these variables, we analyzed eight different experimental condition combinations for each donor, plus a sample of uncultured CD4^+^ T cells (T0) (nine total conditions). Overall, a total of 27 samples were sequenced using the MiniION Oxford Nanopore platform. Sequencing metrics are reported in Appendix A.

To provide an overview of the 27 transcriptome profiles obtained from these three different donors and nine conditions, we first performed a Principal Component Analysis (PCA) (Figure 2). Under the same experimental conditions, PCA revealed that the three donor transcriptional profiles have high similarity to each other except for the PREc samples from the S1 donor (Figure 2A). These samples were thus sequenced individually on separate flow cells, while, due to cost optimization, we decided to multiplex the samples of each donor together for S2 and S3. This batch effect suggests that, if possible, it is desirable to pool together the samples to be compared, even if results were not affected on a functional level.

Focusing separately on the effects of stimulation (Figure 2B), matrix (Figure 2C), and CD4^+^ T cells isolation strategy (Figure 2D), we observed that the transcriptional profile was highly impacted by stimulation. The conditions T0, US, and ST clearly segregated into three different clusters (Figure 2B). Furthermore, while US samples are tightly clustered, the ST samples are more spread out, suggesting that stimulation differently impacts each sample. The impact of the culture matrix appeared to be irrelevant under the US conditions, while we noticed clear segregation of RPMI and PLSM after stimulation, with the RPMI samples more distant from all US conditions (Figure 2C). In this first overview, samples cultured with RPMI showed a stronger stimulation effect. The PCA did not show appreciable differences related to CD4^+^ T cells isolation strategy (Figure 2D).

### 3.2. Identification of Differentially Expressed Genes

To identify differentially expressed genes (DEGs) between conditions, we decided to analyze each donor individually (Figure 3A). Among all possible comparisons (N = 36), we selected the 20 most informative. Genes with an adjusted *p*-value (FDR) < 0.05 and a |logFC| > 1.4 were considered as differentially expressed (Appendix A and Appendix A). Overall, the three donors showed a similar proportion of up-regulated (UP) and down-regulated (DW) genes across the comparisons, with higher numbers of DW than UP genes (Figure 3A). As expected, we found a higher number of DEGs between ST and T0 than between US and T0 conditions both in RPMI and PLSM settings. In accordance with the PCA results, samples stimulated with RPMI showed more DEGs than those found in the PLSM counterpart. Finally, the low number of DEGs between PREc and POSTc was in line with the absence of clustering observed in the PCA.

The intersections of DEGs between ST and US conditions are depicted in Figure 3B. As shown in the figure, non-overlapping DEGs, among donors and matrices, were more frequent than common DEGs. These results suggest that, upon stimulation, different sets of genes were modulated in each sample and that the number of common DEGs across donors was greater in the RPMI vs. PLSM conditions. For example, for the POSTc isolation strategy, we found 106 shared genes exclusively up-regulated after stimulation for donors cultured in RPMI and only 2 with PLSM (Figure 3B).

### 3.3. Functional Phenotype in Different Conditions

To further investigate the functional landscape, we performed a GSEA for a macroscopic view of the biological processes (Figure 4A) and VIPER (Virtual Inference of Protein-activity by Enriched Regulon) analysis to explore the activity of pathways and transcriptional factors (TFs). VIPER allows computational inference of protein activity with two types of input: (1) using gene counts of a single sample or (2) using comparison metrics, such as fold changes or ranking index [13]. We used the first strategy to highlight similarities and differences between individual samples (Figure 4B). Using the ranking index as input, we focused on comparisons between conditions (Figure 5).

Despite the high number of non-overlapping DEGs among donors and matrices after stimulation (Figure 3B), these functional analyses showed a very high concordance between donors (Figure 4 and Figure 6A, Appendix A). These results highlight a strong interindividual variability in terms of DEGs, but also shown a convergence on the same functional mechanisms.

### 3.4. TCR Signaling Cascade before and after Stimulation

As expected, we found that after 18 h of in vitro culture without stimulation, CD4^+^ T cells showed down-regulation of genes involved in immune response processes (Figure 4A, cluster 2). In line with this, the VIPER analysis of pathway activity highlighted that, regardless of the matrix used, the EGFR, p53, TGFβ, and MAPK pathways are inactive in all US samples (Figure 4B). Except for the TGFβ pathway that plays a pleiotropic role in the biology of CD4^+^ T cells [16,17], the EGFR, p53, and MAPK pathways are all involved in T cell activation [18,19,20,21]. Confirming these results, transcription factors (TFs) downstream of these pathways, such as NFATC2 and JUN [22], are also clearly inactive under US conditions (Figure 6A).

Stimulation with PMA/ionomycin bypasses the T cell membrane receptor complex leading to T cell activation, while IL-2 induces the proliferation and survival of TCR-activated T cells [23,24]. After stimulation, GSEA showed the up-regulation of genes mainly involved in processes driven by TCR pathway activation and/or IL-2 stimulation [22], including the immune response to stimulus and cytokine production (Figure 4A, cluster 1 and 3). These results were concordant between plasma and RPMI, but RPMI showed more biological processes with an adjusted *p*-value (FDR) < 0.05.

Pathways expected to be activated by PMA/ionomycin and IL-2, such as NF-kB, TNFα, JAK/STAT, MAPK, and PI3K [23], were indeed found through VIPER analysis in both PLSM or RPMI conditions, and independent of isolation strategy (Figure 4B). Similar to the GSEA results, we found more statistically significant activated pathways with the RPMI conditions than PLSM. To confirm these results, we performed the VIPER analysis on the differences between ST and US conditions (Figure 5). We found that the EGFR, MAPK, PI3K, and NF-kB pathways were significantly activated after stimulation in PLSM and RPMI (Figure 5A and Figure 5B, respectively). Visualizing the interaction networks among pathways and their DEGs after stimulation (Figure 5C,D), we observed a higher number of DEGs involved in these pathways under RPMI conditions compared to PLSM.

Focusing on TF activity, differences between RPMI and PLSM were clearly visible (Figure 6A). In fact, key TFs downstream of the TCR and IL-2 signaling pathways, such as NFATC1, NFATC2, JUN, and REL [23,25], showed more significant activation under RPMI. This suggests that under the RPMI condition, more genes responsive to these TFs were deregulated. Analyzing separately the impact of stimulation in PLSM (Appendix A) or RPMI (Appendix A), similarly to what was observed for the pathways, PLSM and RPMI shared almost the same top 15 active or inactive TFs (Appendix A). However, network analysis showed that the numbers of interactions between TF and DEGs are higher in RPMI than in PLSM after stimulation.

To further confirm the greater activation of the TCR signaling cascade using RPMI instead of PLSM, we compared the normalized counts of genes transcribed by NF-kB1, NFATC2, JUN, STAT5A, and STAT5B, some of the key TFs downstream of TCR activation (Figure 6B). All genes depicted in Figure 6B showed higher levels of expression in RPMI compared to PLSM after stimulation.

### 3.5. Hypoxia and JAK-STAT Pathways Are Activated in CD4^+^ T Cells Cultured in PBMCs

Culturing CD4^+^ T cells as bulk PBMCs and isolating them after in vitro culture (POSTc), we observed that the hypoxia and JAK-STAT pathways were activated in the absence of stimulus (Figure 4B). Conversely, we did not find these pathways in CD4^+^ T cells isolated pre-culture (PREc). These were the main differences between PREc and POSTc isolation strategies. In addition, the hypoxia pathway was active in the PLSM condition after stimulation but was inactive in RPMI.

## 4. Discussion

Human in vitro models can be used for the investigation of individual-specific immune responses. A wide range of techniques has been developed to measure the persistent HIV reservoir in virally suppressed subjects [5,26,27,28,29,30,31,32,33]. These assays can be classified into two groups: (1) PCR/sequencing-based and (2) cell-culture-based assays. PCR/sequencing methods [34,35,36,37,38] include molecular assays recently developed for profiling the transcriptional activity and the chromosomal locations of individual proviruses [38]. Focusing on cell culture assays, many studies of HIV latency and reactivation rely on the evaluation of viral reservoir after in vitro stimulation of PBMCs or CD4^+^ T cells cultured in RPMI (e.g., QVOA, TILDA) [31,33,39,40]. However, a growing body of evidence indicates that different cell culture matrices strongly influence cell behavior in in vitro systems. For example, it is known that serum-free media favors T cells proliferation over media with serum [41]. Additionally, Leney-Green et al. compared RPMI versus human plasma-like medium and found that different concentrations of calcium can strongly impact T cell activation [42]. In this context, we aimed to mimic test conditions that were more representative of patients’ biological samples and decided to explore a more individualized, in vitro cell culture with autologous plasma used as cell culture matrix in comparison to common RPMI. We then considered the possible diverse transcriptional phenotypes of CD4^+^ T cells cultured alone or together with other PBMCs. Human plasma is a rich source of age-specific immunomodulatory components, such as antibodies, complement, cytokines, chemokines, and metabolites [6], which are not usually included in RPMI cell cultures. As examples, we know that activation of TLR pathways [43] that is based on differential cytokine production and that anti-inflammatory molecules (e.g., IL-10) in human plasma [6] may have age-related differences. Moreover, hormone levels, qualitatively and quantitatively different according to sex, can impact leukocyte behaviors [44]. Nonetheless, culture systems in RPMI rarely mimic the plasma concentration of cytokines under chronic disease conditions such as ART-treated HIV infection [45]. It is, thus, not experimentally feasible to recreate age-, gender-, and condition-specific personalization of culture media through supplementation with immunomodulatory agents to account for all such differences.

In line with this, our goal was to evaluate the optimal experimental settings to model the immune response in an in vitro platform upon PMA/ionomycin stimulation through RNA-Sequencing of CD4^+^ T cells exposed to nine experimental settings. We applied a novel single-subject differential analysis approach which is based on the analysis of each donor individually [46,47], conversely to conventional transcriptome analytics strategies that instead require replicate samples to estimate gene-wide data variability and make inferences. Although single-subject methods for identifying DEGs from paired samples need optimization [47], this strategy overcomes a limitation of traditional methods: the cohort/group approaches emphasize the group average rather than individual participants, and this may not represent the individual’s personalized profile. Hence, single-subject methods are needed to dissect the impact of individual genetic and environmental variability on transcriptomic profiles and, thus, are crucial tools in precision medicine [47]. With this approach, we observed that despite a high proportion of DEGs after stimulation among donors and culture matrices (Figure 3B), functional analyses (Figure 4 and Figure 6) revealed strong biologic concordance.

In this work, to simulate in vitro T cell activation, we used a combination of PMA/ionomycin and IL-2, as already described in the TILDA enhanced assay [48]. PMA activates protein kinase C (PKC), which induces the IkB kinase (IKK) and Ras/Raf/MAPK signaling pathways, leading to the activation of both NF-kB and AP-1, respectively [23]. Ionomycin is a Ca^2+^ mobilizing agent that induces calcineurin-mediated dephosphorylation and import of TFs from the NFAT family into the nucleus [23,49]. Together, PMA and ionomycin provide stimulation allowing bypass of the TCR activation, resulting in T cell activation with the production of a variety of cytokines and in the reactivation of HIV proviruses in infected cells. This is because HIV transcription is positively regulated through binding to the promoter of NF-KB, NFAT, in particular members of NFATc1 and NFATc2, and AP-1 (activating protein-1), a dimeric transcription factor composed of JUN, FOS, or ATF (activating transcription factor) [50,51,52,53]. IL-2 was used because it induces the proliferation and survival of TCR-activated T cells activating the STAT TFs family, in particular STAT5, STA3, and STAT1 members through the JAK/STAT pathway [24].

Our results show stronger activation of the TCR signaling cascade when RPMI rather than autologous plasma is used. In particular, levels of key genes that are known to be expressed by TFs downstream of this cascade, such as IL-2 and TNF, are clearly higher in RPMI after stimulation compared to plasma. These results suggest that autologous plasma contains factors that may provide negative feedback to constrain TCR signaling cascade activation. Previous studies compared IMDM (Iscove’s modified Dulbecco’s medium) [54] and HPLM (human plasma-like medium) [42] to RPMI under TCR stimulation conditions. Both studies showed that the lower Ca^2+^ concentration in RPMI limits maximal ionomycin stimulation. Despite this lower Ca^2+^ concentration, RPMI allowed stronger T cell activation compared to autologous plasma, but it is likely that this high level of in vitro activation does not reflect the magnitude of in vivo activation. This could lead to an overestimation of mechanisms studied in vitro, such as HIV reactivation. At the same time, use of complex media such as plasma may obscure biological results due to inter-individual variability.

Concerning the differences between CD4^+^ T cells cultured alone (PREc) and CD4^+^ T cells cultured with other PBMCs (POSTc), we did not find after 18 h relevant differences in the stimulated samples. Nevertheless, the unstimulated samples showed activation of the hypoxia and JAK-STAT pathways in CD4^+^ T cells cultured as bulk PBMCs and isolated after in vitro culture (POSTc), but not when cultured alone (PREc). The JAK-STAT pathway is a crucial transmitting signal cascade from many cytokines and growth factors into the nucleus, regulating a number of cellular functions such as proliferation, inflammation, immune response, and angiogenesis [55,56]. JAK-STAT activation is triggered by cytokine receptors, including interleukin (IL) receptors and interferon (IFN) receptors, so cellular communications that rely on cytokine signals produced by other PMBCs in culture could explain this result [55,57]. Furthermore, the JAK/STAT pathway is known to be closely related to hypoxia [58,59,60,61,62]. We hypothesize that the other PBMCs act as oxygen biosensors and, through molecular signaling, induce the activation of these pathways in CD4^+^ T cells.

## 5. Conclusions

In conclusion, this study provides insights into the CD4^+^ T cells transcriptional landscape under nine different experimental settings. Despite several studies focusing on HIV reactivation evaluated distinct CD4^+^ culture conditions [63,64,65], to our knowledge, this is the first work that compares the CD4^+^ transcriptional profiles before and after the in vitro culture and stimulation, simultaneously benchmarking two matrices, and that analyzed the impact of culturing CD4^+^ alone or with other PBMCs. Exploring all these experimental settings in the same individual could provide useful knowledge both to evaluate the optimal cultural conditions and to consider the difference in adopting one experimental setting or another. This in-depth analysis showed that isolation timing did not mainly affect results, while RPMI induced a stronger T cell activation as compared to PLSM. Such results were both confirmed by functional analysis (enrichment of biological processes and inferred activity of pathways and transcriptional factors) and by the higher expression of genes downstream of the TCR cascade. Based on our results, we hypothesize that in vitro assays based on simpler matrices (as RPMI) could reduce the complexity and simplify results evaluation, highlighting the qualitative aspects of an in vitro cell culture, whereas the selection of autologous plasma as a matrix could provide more realistic findings for studies aiming at quantifying the magnitude of a biological mechanism under TCR cascade activation. Thus, we believe that this gap found between RPMI and PLSM cell culture results could help researchers to choose the optimal in vitro culture conditions based on their scientific aim.

## 6. Limitations of the Study

Although we applied an in-silico inference approach to impute the pathways and transcription factors activity, our analysis was restricted to the transcriptome profile of T lymphocytes rather than a direct investigation of the molecular and metabolic state of primary cells under the nine experimental settings. Further, the low frequencies of infected CD4^+^ T cells in early-ART-treated patients did not allow us to measure HIV reactivation after in vitro stimulation. Moreover, low sequencing coverage limited the study to only the more expressed transcripts. Lastly, although we observed a surprising biological concordance between donors (*n* = 3), it would be desirable to replicate these experiments on a greater number of individuals including healthy donors.

## Figures and Tables

**Figure 1 biomedicines-11-00888-f001:**
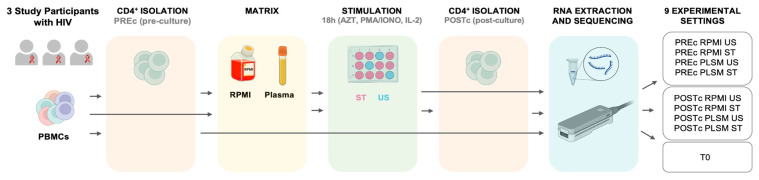
Experimental design depicting the sequence followed to investigate the expression of the CD4^+^ T cells genes from 3 young adult donors with perinatal HIV and in 9 different experimental settings. Lymphocytes were isolated from PMBCs before or after culture (respectively, PREc and POSTc) and stimulated in the presence of PMA/ionomycin, AZT, and IL-2 (ST), or left unstimulated for 18 h (US) in plasma (PLSM) or RPMI. An additional sample of uncultured CD4^+^ T cells (T0) was collected as a transcriptional basal state. Total RNA was extracted and sequenced using Oxford Nanopore technology.

**Figure 2 biomedicines-11-00888-f002:**
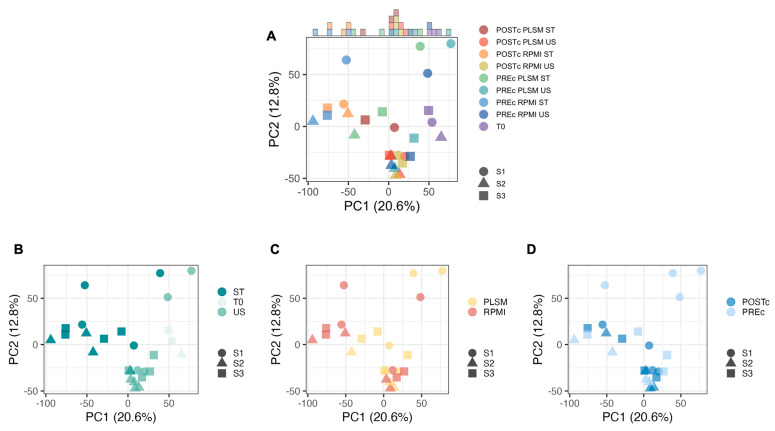
Overview of the transcriptomic profiles of the 3 study participants (S1, S2, and S3) in the 9 experimental settings analyzed through Principal Component Analysis (PCA). Rectangles on the top indicate the frequency of each experimental setting along the *x*-axis (PC1). (**A**). To highlight the impact of single experimental conditions, the same PCA was colored according to stimulation (**B**), culture medium (**C**), and CD4^+^ T cells isolation strategy (**D**).

**Figure 3 biomedicines-11-00888-f003:**
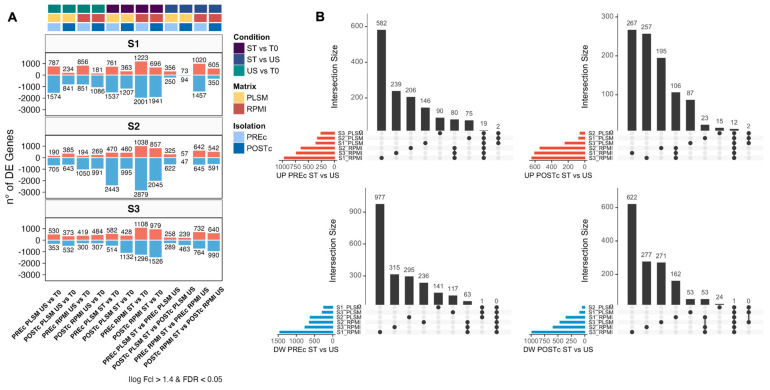
Differential gene expression analysis. Number of genes up-regulated (red) and down-regulated (blue) among US vs. T0, ST vs. T0, and ST vs. US comparisons, analyzed in the 3 study participants (S1, S2, and S3) in both PREc and POSTc. Genes with |log fold change| > 1.4 and adjusted *p*-value (FDR) < 0.05 were considered differentially expressed (DEGs) (**A**). Overlap of genes up-regulated (UP, red) or down-regulated (DW, blue) after stimulation among donors and matrices for CD4^+^ T cells isolated before (PREc) or after (POSTc) culture (**B**).

**Figure 4 biomedicines-11-00888-f004:**
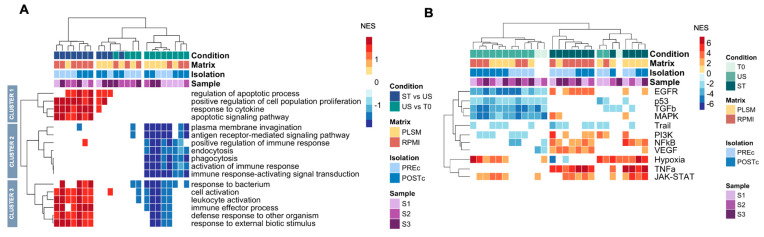
GSEA on Biological processes and inference of the pathway activity. Heatmap shows Normalized Enriched Score (NES) values from GSEA on biological processes between ST vs. US and US vs. T0 comparisons. Positive values (red) indicate biological processes enriched in up-regulated genes. Conversely, negative values (blue) show processes enriched in down-regulated genes. White indicates the absence of significant results (**A**). VIPER analysis to infer pathway activity in a single sample’s transcriptome. Active pathways are shown in red, inactive pathways are shown in blue, and pathways with no significant results are shown in white (**B**).

**Figure 5 biomedicines-11-00888-f005:**
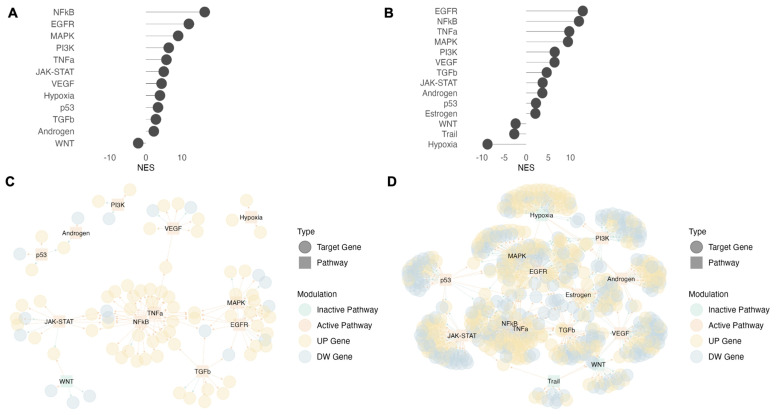
Differentially activated pathways between ST and US conditions using PLSM (**A**) and RPMI (**B**). Networks connecting pathways activated (orange rectangle) or repressed (green rectangle) after stimulation with the relative up-regulated (yellow circle) or down-regulated (blue circle) genes both in PLSM (**C**) and in RPMI (**D**). The connections between pathways and DEGs nodes and their mode of regulation (positive in orange or negative in green) are information from the PROGENy database.

**Figure 6 biomedicines-11-00888-f006:**
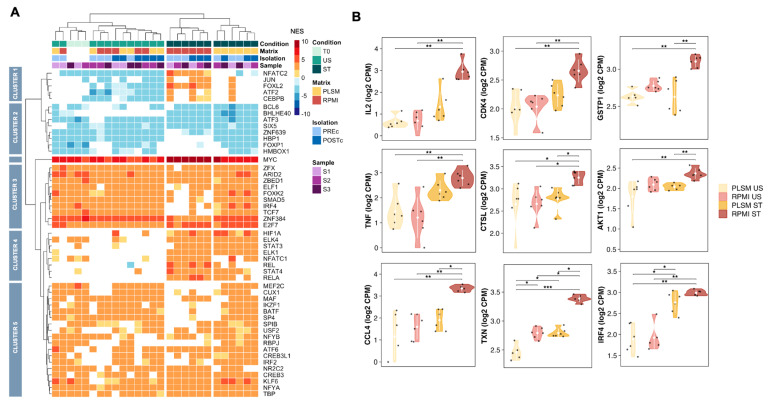
Inference of transcription factor activity and expression of their target genes. Heatmap of NES values from VIPER analysis on the transcription factor (TF) activity of single samples. Positive values (red) indicate active TFs, negative values (blue) show inactive TFs, while white indicates no significant results (**A**). Violin plots showing the expression difference after stimulation both in PLSM and in RPMI conditions of selected genes regulated by the NF-kB1, NFATC2, JUN, STAT5A, and STAT5B TFs. Statistical significance is indicated as follows: * for p-values less than 0.05, ** for p-values less than 0.01, and *** for *p*-values less than 0.001 (**B**).

## Data Availability

The RNA-Seq data from this study were uploaded to GEO and may be downloaded using the GSE224135 accession number.

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
