# Peer review of "How CD4+ T Cells Transcriptional Profile Is Affected by Culture Conditions: Towards the Design of Optimal In Vitro HIV Reactivation Assays"

_biomedicines, 2023, doi:10.3390/biomedicines11030888_

Round 1

Reviewer 1 Report

"Impact of media culture and PBMCs signaling on CD4+ T cells transcriptional dynamics to inform the optimal design of an in-vitro HIV reactivation assay". The manuscript by Pascucci et al. evaluates the effect of plasma on transcriptional profiles of CD4+ T-lymphocytes. Although the study addresses the important component limitations of in vitro conditions, it has a few limitations.

Overall, all the figures' quality is compromised, making it impossible to read the labels in the images.

Authors should include secondary validation at protein levels of a few markers of the signaling networks. 

Various studies have been published about the culture conditions of PBMCs and their effect on HIV replication. What is the difference in the current study? The authors should clearly state the limitations of related studies and how the present study addresses those limitations. 

Did the authors observe any effect on the viability of the cells in different conditions? It will be nice to have that data.

Authors should include a table of DEGs in the supplementary materials. Also, the data availability statement does not have the link/address.

Minor: Line #72 Within first year of life? Should be within first year of infection

Reviewer 2 Report

Dear editor in chief

The manuscript entitled: “Impact of media culture and PBMCs signalling on CD4+ T cells transcriptional dynamics to inform the optimal design of an in-vitro HIV reactivation assay” addresses a highly interesting topic. The present study is aimed to determine the impact of selecting this artificial medium, largely employed in research, in comparison to human plasma and how the choice of culturing and stimulating a single lymphocyte population (CD4+ T cells) as opposed to bulk PBMCs influences CD4+ T cell gene expression profiles. The article is well designed and well written. I suggest that the authors could greatly enhance the usefulness of this work for the community if they consider the following changes:

1-      Authors should explain each of the abbreviations the first time it appears in the main text. Also, it is better to provide a list of abbreviations.

2-      The title needs to be improved. “Impact of media culture”???

3-      The Figures are very small and should be presented with appropriate resolution and magnification. Also, if there is no suitable space in the text, it is better to use them in the supplement.

Round 2

Reviewer 1 Report

I thank the authors for their detailed responses and revisions. The figure quality and the manuscript overall are of much better quality.